# Older People’s Lived Perspectives of Social Isolation during the First Wave of the COVID-19 Pandemic in Italy

**DOI:** 10.3390/ijerph182211832

**Published:** 2021-11-11

**Authors:** Sabrina Cipolletta, Francesca Gris

**Affiliations:** Department of General Psychology, University of Padua, 35131 Padua, Italy; francesca.gris@studenti.unipd.it

**Keywords:** coronavirus, COVID-19 pandemic, loneliness, information and communication technologies, older people, qualitative method, social isolation, social support

## Abstract

The aim of the present study is to understand the experiences of isolation and strategies used to cope with it among older people living at home during the first wave of the COVID-19 pandemic. More specifically, the roles of media and online technologies were also explored. Semistructured interviews were conducted via telephone between March and April 2020 with 30 people aged 72–94 years old living in Northern Italy. The thematic analysis identified six thematic areas: changes in daily life, emotions, social networks, exploited resources and strategies, use of media, and view of the future. Older people faced the emergency in heterogeneous ways; some were able to take advantage of their own residual resources and of social support, whereas in other cases, isolation exacerbated existing weaknesses. Technology and media were useful for reducing loneliness and fostering social contacts, but people with age-related impairments or low digital literacy presented many difficulties in approaching new technologies. Moreover, the overabundance of information could also increase anxiety and feelings of threat. Given the impact of social isolation on older people’s well-being, it is critical to identify and strengthen personal resources and social support strategies that may help older people cope with the restrictions imposed by the COVID-19 pandemic.

## 1. Introduction

The COVID-19 crisis was defined as a global pandemic in March 2020 by the World Health Organization [1], a decision that urged governments to establish anti-infection measures such as hygiene practices, social distancing, and lockdowns, thus imposing drastic changes in people’s everyday routines and lifestyles [2]. Italy was the first Western country to be greatly affected by the pandemic and in which strict preventive measures were applied. The entire population had to remain at home except in cases of extreme need and could not visit people outside their own households, with consequences for their well-being [3]. In addition to the distress of the lockdown was added a sense of threat caused by a dangerous but invisible virus [4,5,6].

Since the very beginning of the pandemic, authors [7,8] have claimed that older people (aged 60 and over), especially those already suffering from a chronic disease, were at a high risk of contracting the virus and that the COVID-19 pandemic could be considered a geriatric health emergency [9]. Isolation and restrictive practices were implemented to protect older people from infection, but these restrictions had negative impacts on their health and well-being [10,11,12]. This has been defined as the social connectivity paradox, as social distancing improves protection against COVID-19 exposure but also increases isolation, loneliness, and depression [13].

Previous studies have suggested that social isolation among older adults increases the risk of psychological and medical problems [14,15], particularly depression, anxiety [16], and poorer sleep quality [17]. Recent studies [18] have pointed out that loneliness increases the emergence of post-traumatic symptomatology and, more importantly, that both received and perceived social support could be protective factors against psychological distress [19]. Based on these results, many authors [20,21,22,23,24] have proposed strategies to support older people by maintaining social support and reducing loneliness.

New and traditional technologies (i.e., telephones, smartphones, computers) could be used to provide social support and increase their sense of belonging [25,26,27]. Moreover, psychological counselling and therapy could be delivered online [28,29]. However, the engagement of technology could present some problems. First, disparities exist in access to smartphones or internet services for many community-dwelling older adults [30]; second, there is a literacy divide in digital resources and a lack of technology skills [14], especially for those who have lower educational levels and socioeconomic statuses [10]. Some researchers [31] have already suggested encouraging younger generations with higher technical skills to support older people in the use of new media. Traditional media (e.g., television and radio) can also be useful instruments to maintain connectivity with the rest of the world. However, exposure to news could cause distress and more anxiety [10], so it is recommended to regulate the use of media. It has been suggested that people should limit their time searching for information—a maximum of once or twice per day [17] and preferably not at night, to facilitate falling asleep [32]. During the COVID-19 pandemic, people were bombarded with a large amount of information, which authors [33] have defined as an emerging infodemic that has a negative impact on people’s health.

To date, no study has qualitatively analysed Italian older people’s experiences of the pandemic and their use of new and traditional media to cope with isolation. A useful framework to understand this experience is the personal construct theory (PCT), which was first introduced by George Kelly [34]. PCT is a theory of personality that allows for a deep and coherent psychological understanding of the person experiencing life events and challenges [35]. Kelly [34] identifies some transitions, which are diagnostic constructs used to define the meanings that people give to changes in their own construction systems. Transitions have been previously used to understand the illness experience [26,36] and the present pandemic [4,5,37].

The first transition usually experienced when faced with the pandemic is anxiety because it represents an unknown and unprecedented situation that cannot be construed within a personal construct system. Kelly [34] described the transition of anxiety as a situation in which an event—usually experienced as unknown or unprecedented—cannot be construed within an individual’s personal construct system. Other people can experience the threat of an imminent comprehensive change to core structures, those central to one’s identity and in PCT terms threat is represented by experiences that endanger one’s identity, certainties and values [34]. The transition of guilt is defined as dislodgement from one’s core role, one’s characteristic way of being [34]. The person may cope with these transitions through different strategies [5]. One is represented by constriction, defined as the reduction of one’s perceptual field in order to minimize apparent incompatibilities in construing. Another is hostility, defined as the attempt to extort evidence in favour of a type of social prediction that has already been recognised as a failure. Finally, people may actively elaborate the situation and make the best out of it. The active elaboration of one’s own perceptual field is defined by Kelly [34] “aggression”.

The present study aimed to explore the perception of isolation on the part of older people living at home during the first lockdown experienced in a Western country due to the COVID-19 pandemic. Moreover, this study intends to help understand participants’ opinions, feelings, perceived changes in everyday life, and strategies used to cope with isolation. More specifically, the roles of media and online technologies were also explored. Participants’ experiences were differentiated based on the transitions elaborated within PCT to identify personalized strategies for health promotion.

## 2. Materials and Methods

A qualitative approach was implemented in our study as the gold standard to explore and deepen people’s life experiences through a storytelling process aiming for understanding [38].

### 2.1. Sampling and Recruitment

The present study involved older people from an area of Northern Italy, where containment measures were implemented earlier and more strictly than in any other Western country. Inclusion criteria were being older than 65 years, living at home, and being able to understand the questions and answer in an appropriate way.

Participants were recruited by researchers thanks to their collaboration with a charity association (the Saint Egidio’s community) and by snowball sampling. This method consists of starting from a group of known individuals and then increasing the sample based on the network of participants [39]. According to the criteria for qualitative research, sampling ended once theoretical saturation was reached, which is the point at which gathering more data does not lead to more information related to the research questions [40], usually when the number of interviews reaches around 15 (+/−10) [41]. The final number of participants was 30. Two interviews were excluded from the analysis; one participant decided to withdraw from the study and the other interview was incomplete. The sample analysis is thus composed of 28 participants (10 M, 18 F) from 72 to 94 years old (mean = 80, SD = 5.742), all Caucasian and retired with a low (N = 12), middle (N = 5) or high (N = 11) socio-economic status, married (N = 14), widowed (N = 11), or unmarried (N = 3), and living at their homes alone (N = 12), in a two-person family unit (N = 14), or with a home health aide (N = 2).

The Ethics Committee of the Human Inspired Technology Centre of the University of Padova approved this study. Participation was voluntary, and participants received a letter containing all the information regarding the study. During the first contact, participants gave their oral informed consent to participate in the study (which was recorded before the interview started) and subsequently filled in the informed consent form left in their mailboxes. Researchers behaved respectfully towards participants, and special attention was given to preserving participants’ anonymity by the use of codes in the text.

### 2.2. Data Collection and Analysis

A doctor in psychology, trained to conduct semistructured interviews [41] and with previous experience of working with older people, conducted the interviews through telephone calls between the 27th of March and 15th of April 2020, a period when the region was in total lockdown. The average duration of the interviews was 23 min, with most of the interviews lasting between 30 and 40 min and only a few around 15 min. The interviewer chose to respect the wishes of those participants who preferred to have shorter calls because they were not comfortable spending a longer time on the telephone, and these interviews ended when participants had nothing to add to their given answers. These shorter interviews were not excluded from the analysis because they were informative of a particular way of coping with isolation. Although flexible, the interviews followed a common guide comprised of the following questions: How are you experiencing this period? What differences do you see compared to before? What solutions have you identified to deal with the situation? With whom are you in contact? How do you keep in touch with other people now? Is there anyone you cannot keep in touch with because of the pandemic? Which technologies do you use? Which technologies do you prefer? Do you have something else to say?

Interviews were registered and transcribed verbatim. The qualitative analysis followed the five steps that Braun and Clarke [42] suggest for thematic analysis: (1) familiarization with the data; (2) generating initial codes; (3) collecting similar codes in overarching themes; (4) reviewing themes; (5) refining and naming themes. Finally, themes were combined to group together similar experiences based on the transitions elaborated within PCT.

The study was conducted according to the Consolidated Criteria for Reporting Qualitative Research checklist [43]. Coherence and reliability were achieved by accurately reporting interview administration, data analysis, and consistent use of each interview’s quotations. Reflexivity was sought through repeated comparison of the themes concerning the data, and discussions were held between the researchers about alternative interpretations of the results [44].

## 3. Results

The data analysis identified 47 themes grouped in six thematic areas, as listed in Table 1. The changes in daily life referred to participants’ difficulties and attempts to adapt to the changes in their habits and other aspects of their lives due to the restrictions imposed to contain the COVID-19 contagion. The perceptions of these changes varied among participants and influenced their emotions. Themes relating to social networks show the characteristics of participants’ social relationships that influenced the possibility of taking advantage of internal or external resources to cope with the situation. The thematic area of the use of media involved both perceptions of efficacy or uselessness of technology and knowledge or failure in the use of these instruments. All participants noted an increase in media use to stay connected with their social networks; nevertheless, not all of them took advantage of this opportunity in the same way. Finally, participants spontaneously proposed different views of the future derived from the changes they had experienced and the strategies found to cope with them.

The relations between the themes are illustrated in Figure 1. The core category underlying the themes was social isolation, but participants dealt with it in varying ways, as pointed out in the description of each theme. The transitions of anxiety, threat, guilt, constriction, and aggression—as defined within the PCT approach—were used to describe these diverse ways of coping with the common experience of isolation. Each transition is described through the presentation of the experiences grouped together on each theme: changes in daily life, emotions, social networks, exploited resources and strategies, use of media, and view of the future. When the themes or the codes attributed to a specific experience characterizing a certain transition are not explicitly mentioned in the text, they will be reported in parentheses.

### 3.1. Anxiety

Anxiety was the main transition experienced when participants found it difficult to give meaning to the pandemic, which turned their lives upside down to the point of making them unrecognizable. Changes brought on by the pandemic were described as unreal and belonging to another world. The narratives themselves were often difficult to follow and seemed chaotic to the interviewer: “I never expected to live something like this [the pandemic] into my life” (P3, F, 73); “I seem to live in another world” (P10, F, 88); “In certain moments it looks like if I was on another planet, I seem to dream because I say to myself: ‘can it be possible a thing like this?’ […] Sometimes I don’t know if it is reality or fantasy” (P26, F, 94).

These participants’ prevalent emotions were incredulity about the spread of the virus and the fear of confronting something unknown and invisible. This feeling also escalated to panic and obsession: “When you come home, you never know if it [the virus] is there. […] When I come home, I start to disinfect myself, wash myself, disinfect everything, everything that comes into the house and so [there] is a little bit of an obsession” (P3, F, 73).

Not being able to give meaning to their own experience, interviewees implemented strategies that proved to be inefficient in coping with anxiety, or they looked for a possible origin of the virus in supernatural entities (exploited resources and strategies): “I can’t read, I can’t sew, because I can’t see [what this is]. This is not living” (P8, F, 83); “I pray three times a day against this thing that is all around. […] Because this thing [the corona virus] comes from Satan […] Who punishes us is Satan” (P27, F, 84).

Narrations showed that these participants passed from the panic of going out of their home to going around to escape from loneliness: “I don’t go inside the houses, but while going through the street I go to ring the bell and I say, ‘Hello, how are you? Is it all right?’” (P8, F, 74). One participant became suspicious about other people who were viewed as possible vehicles of infection and felt angry at people who denied the situation: “When I see some people who are not taking it seriously, they scare me, they make me [so] angry that you have no idea” (P3, F, 73).

These participants were alone or had wide undifferentiated social networks. Other people’s behaviours were described as unknown and unpredictable: “I do not know when the path will be given to Liberation, that is when people will be able to go out, here and there, what might happen … because people stay at home. Yes, they do but also, they do not … my sister does not respond to the phone, does not hear, does not respond. To be honest, I don’t even know where she lives” (P27, F, 84).

Sometimes participants maintained their pre-existing relationships through the telephone but this could become an additional source of anxiety, either due the difficulty in using it or due to receiving a large number of WhatsApp messages: “[How to use the smartphone] is all to be learned but there is not much passion to learn it, this is the trouble, because I feel powerless and I am tempted to throw it away […] We [I and my friends] message each other even too much because […] you open your cell phone and find 110 messages, I say ‘it is not possible’” (P3, F, 73). Television also might become a source of anxiety for the overwhelming news transmitted: “When they [media] show you how many people are dead, how many of these or those [things], I said […] ‘Stop now otherwise you go out of your mind’” (P8, F, 83).

Views of the future were vague, and participants pointed out the difficulty of anticipating it: “I don’t know how this is going, will we get ill too? I don’t know” (P7, F, 74).

### 3.2. Threat

Some participants were mainly threatened by the possibility that they or their loved ones might contract the virus: “This period has begun with a certain fright […] Also because my husband and I are of an age to be at higher risk” (P5, F, 74). Changes undermining their certainties were perceived as pervasive and involved the physical domain in terms of sleep, appetite, and attention disorders: “What is the problem? I don’t eat at the right times at home, and I don’t go out but I’m always deadly hungry” (P9, M, 83). Concern was the prevalent emotion, and led these participants to stay at home and rigorously follow anti-contagion measures with the consequent limitations in their daily life and social relationships because others were seen as possible vehicles of contagion: “ I respect the rules I have received” (P12, F, 74); “I always go out with the mask on, gloves and mask” (P4, M, 89); “My son lives here, in the same floor […] but we don’t see each other because they go out. Maybe they bring something home” (P7, F, 74).

In terms of social networks, these participants usually had few strong social ties (mainly within the family) and used to rely on them to solve practical problems or emotional issues: “It is [our daughter] who brings us our groceries home. [...] She brings us our groceries, she leaves them in the street in a basket” (P9, M, 83); “Luckily, I have my husband, and so we exchange a few words” (P22, F, 80). They kept in touch with neighbours if there was already a pre-existing relationship; in this case, people with bordering backyards felt they had an advantage because they could keep in contact with others without being too far away from their home: “You stay in the garden and you have a chat [...] my contacts are only with my neighbours and we cannot do anything else, when I go out with my dog I always meet the same people I met before with their dogs, and we say goodbye, we have no contact” (P21, F, 77) “We are all inside our houses. Nobody goes out, nobody speaks to each other, no one opens the balcony window to allow the air in. You see nobody around and I have no relationship with anyone because most of my neighbours are new. I had a good relationship with the lovely people who lived here before, but they are not here anymore. Therefore, I don’t know these new ones” (P20, M, 84).

The only medium used to keep in contact with a close circle of relationships, yet also at a distance, was the telephone, whereas using new media was felt to be tiring especially as it was considered something new that might imply a change in the participants’ comfort zone: “Unfortunately I struggle to use those very modern things... I have only the telephone. I only make and receive calls” (P4, M, 89). Due to worries about the pandemic, the participants watched television to keep themselves “informed” (P6, F, 74) but exposure to lots of bad news ended up becoming a source of additional worry due to the huge amount of alarming news broadcast: “I watch the television to catch up on the pandemic” (P20, M, 84); “I watch the television, what they say, but not too much because otherwise, it makes you sick” (P24, F, 72).

Finally, threat involved participants’ views of the future that were characterized by concerns about the psychological consequences of isolation: “The sooner [the isolation] ends, the better. Otherwise, if it goes on, they will have not only the virus but also depression” (P22, F, 79).

### 3.3. Guilt

Due to the pandemic, ten participants experienced changes with a loss of their personal roles (guilt) and felt a gap between their previous lives and present conditions. These participants were used to mainly looking after themselves and taking care of others (not just family members but also friends and other people in the community) rather than asking others for help. The pandemic introduced profound changes in these core aspects of their lives. In particular, participants complained about the loss of their role as grandparents: “I took care of my grandchildren, I took them, I brought them, and so on” (P1, F, 78). Moreover, narrations showed that participants remained at home as much as possible because they understood the importance of norms and felt responsible for others’ safety: “I am duty-bound, that is, if they impose something on me, I think about it and I do it. [...] I am very responsible and so is my husband” (P12, F, 74).

Although acknowledging the existence of the virus and understanding its risk, three participants considered isolation as a limitation that caused a loss in their activity (e.g., travelling, physical activity) while ageing: “I was and am a person who goes out at 9 am and comes back at 1 pm; who goes out again at 4–4.30 pm and comes back at 8 pm. Home makes me feel nostalgic […] I feel stuck in writing, stuck with my whole nervous system. I always feel nervous, alone […] Before, one was free; now we are [in a] home detention agreement without having committed any crime” (P13, M, 79).

The deprivation of physical contact, limitations in their lives and in the possibility of taking care of others led to a loss of meaning and caused feelings of loneliness and sadness, which were the prevalent emotions expressed in these participants’ narratives: “I miss physical contact with other people who we now only talk to by phone. […] I have to hold them [my grandchildren] off, this is a sorrow” (P12, F, 74); “I hear the bells but I can’t be in the church, because the churches are all closed. I need this and I want to die” (P26, F, 94); “I call my daughter very little because she works at the hospital. [...] She hardly calls me, I am reluctant to disturb her” (P1, F, 78).

Some of these participants watched television to cope with loneliness, because it was an already widely used strategy to distract themselves without being a burden to others. At the same time, someone emphasized the spread of information linked to the pandemic, which was usually expressed in a crude and repetitive way: “Many people said to me that they switch off the television because it makes them crazy, they said that it conveys terrible anxiety, distress, and depression” (P1, F, 78).

These participants did not often speak of the future, but when they did, it was in a pessimistic way, heightening the feeling that nothing would be as it was before: “If we will get out, when we will get out, I don’t know, and how we will get out, but we will be a bit suspicious of each other” (P3, F, 73).

### 3.4. Constriction

In the face of anxiety, threat, or guilt, seven participants tried to maintain or regain their usual constructions of themselves and the world by narrowing their perceptive fields (constriction) and excluding the new elements introduced by the unprecedented situation represented by the COVID-19 pandemic. Some of these participants did not recognize any changes between their present and previous lives: most were already used to staying at home due to age-related health problems or caregiving roles. In one case, a method to continuing their previous way of life was to find subterfuges to dismiss the restriction norms: “When I was younger, I liked to go out. Now I stay quiet because […] I have a lot of illnesses” (P2, F, 88); “I have to take care of my ill husband, so I’m still at home” (P15, F, 81); “I have to sneak around to go into the woods and not be seen” (P14, M, 76).

Constriction resulted in a lack of expressed emotions and the attempt to avoid answering questions regarding feelings associated with the current situation: “I am not living badly, eh, because, you know, at my age, it’s not that it is so hard to stay at home” (P2, F, 88); “I live well enough that I have no particular problems. […] I do what is allowed, for example, the groceries. I just do this. Do you have any specific questions?” (P11, M, 81).

In contrast, some participants stressed their condition of loneliness, which often continued from pre-pandemic times, as they did not have many active social relationships: “I love staying alone” (P14, M, 76); “My wife died. [...] Now this disease, this virus, it happened just a month ago when I fell down and I broke my rib and I had to stay home for 30 days all alone, always, I repeat—all alone” (P28, M, 86).

Social isolation was increased by the lack of using new technologies, which in turn was due to a lack of skills or the will to use them to keep in contact with others: “I have never had contact with anyone. Not with the telephone, or with other media” (P28, M, 86). Age-related impairments could make it hard to learn new applications, such as sending a message, or reading it if it had a small font size, or remembering a telephone number by heart: “I have a problem with my eyes and I see very little [...] if my friends send me a message, I don’t see it” (P16, F, 88).

These limitations provided a sense of frustration, but participants avoided this feeling by attributing their sense of inadequacy to the technology. In addition, one participant who had good skills still considered these media useless because they lacked all the aspects of face-to-face communication: “They aren’t like real contacts. When you talk with someone and you look them in the face you can see their reactions; when you talk with someone on the computer, you don’t have this chance” (P11, M, 81).

The narrowing of these participants’ life perspectives also resulted in a lack of future perspectives, expressed by the absence of any references to the future.

### 3.5. Aggression

Eight participants were not overwhelmed by the events and instead changed their own constructions to adapt to the new situation (aggression). These people reorganized their everyday lives around allowed activities: they found new solutions to cope with the situation (e.g., one person started making protective masks and another listened to the mass on television) or took isolation as an opportunity to carry on or rediscover their hobbies and passions or discover new ones and reported positive feelings associated with these activities: “I live in a very peaceful way. […] I’ve discovered the pleasure of reading” (P19, M, 80).

They reported that feeling committed helped them to cope with loneliness and other negative emotions: “I spend the afternoon praying, then I turn on the record player and listen to music. […] Then I do crossword puzzles or if it’s sunny, I stay in the garden. […] I try to fill the day with these activities” (P6, F, 74); “I also have a lot to do at home. I do all the jobs I have never done because I had no time” (P7, F, 74); “The cleaning lady no longer comes since it began. […] So I have to provide for myself” (P20, M, 84).

Some participants also attributed their positive feelings to how their character complemented the ongoing situation: “In a certain sense, this is a strange thing because I have always been a lover of peacefulness and to see the streets so empty gives me a sensation of -bah, sometimes not seeing crowds of people makes me feel better” (P9, M, 83).

Although they suffered from being distanced from loved ones (social networks) and with limitations in their daily activities, these participants respected the restrictions because they considered them necessary to protect the community’s health. They were usually attentive to other people’s needs and to the common well-being: “In this period, I am obviously at home; I go out very little, only once every 15 days for the groceries” (P17, M, 81); “I wish that […] we can take the chance to change. I have lived most of my life but I think about the world that we’ll leave to younger people. We have to leave them a different world to the one we live in now, which is one we have broken (P5, F, 73).

In general, they autonomously provided for their needs but could also rely on others (community services, home health aides, and children) for help: “I went to my car the other night and I found [the battery] dead […] Then a gentleman who lives nearby came into my courtyard with his car and we have worked it all” (P9, M, 83).

These participants maintained their relationships with relatives, friends, neighbours, and colleagues through the use of technology: “Luckily, during the rest of the day, I can have indirect, but useful, contact through the smartphone with both children and grandchildren” (P20, M, 84).

These tools were considered as good alternatives to the usual modalities of maintaining relationships. One person was already skilled in the use of new technologies due to personal interest or previous use at work and was used to maintaining contact with others through them: “I keep in touch with others through the telephone or an email. [...] I used them even before, when in 1995, the Internet came out and it seemed a positive thing” (P18, M, 76).

Other participants, while starting from low levels of competence, were motivated by the situation to rely on technology and took the opportunity to improve their ability: “For Christmas, I’ve received a smartphone, and so I have a way to send messages, to send videos, send beautiful things to hear that come to me every day. Wonderful things to watch come to me. And this helps me a lot” (P24, F, 72).

Some of these participants looked at the end of the pandemic as an event able to modify people’s moral values and to make humanity better (view of the future); they compared the pandemic with living through a world war, after which every little thing was appreciated: “I hope, I say I hope, that when this emergency is over, people have a different way of seeing life. That it is a sign, a sign for how you lived before and after, as [during] the post-war” (P6, F, 74).

## 4. Discussion

This study aimed to explore how older people experienced and coped with isolation during the lockdown in the first Western area affected by COVID-19. The results reflected the particular context in which they were collected. In February 2020, the first cases of people impacted by the COVID-19 were registered in Italy. By the time data collection had begun (i.e., 26 March), there were 80,593 total cases and 8241 deaths in this country [45] and the population had been under a complete lockdown for 20 days. People had to remain at home and to wear masks and gloves at the supermarket, and it was strongly recommended not to go to the supermarket more than twice a month or once a week. The present study involved older people living in Veneto (an area of Northern Italy), where the containing measures were implemented earlier and in a harder way than in any other Western country [46] as this region registered one of the highest numbers of infections (6935 total cases on 26 March) and one of the highest numbers of bed occupancy in Intensive Care Units (326) [47].

The results of the present study highlight a variety of responses in terms of day-to-day life changes, emotions felt, social networks, exploited resources, and the use of media. Previous studies have already underlined that older people form a large and heterogeneous group; for example, they differ in terms of genetics, life and cultural experiences, health, and lifestyle [48]. To better describe the heterogeneity of the answers collected and identify common trajectories, the researchers referred to the main transitions in terms of PCT that participants experienced concerning the changes brought by the COVID-19 pandemic: anxiety, threat, guilt, constriction, and aggression.

The awareness that one can construe, only partially, events that are on the border of the range of convenience of one’s construct system is what Kelly [34] referred to as anxiety. In line with previous studies [5,16], anxiety in particular was the prevalent experience for some participants in this study who spoke about the pandemic mainly in terms of something invisible and unknown that upset their lives. They could not find an effective way to cope with the situation. Instead, they resorted to superstitious solutions and became suspicious of others and the outside world, thereby living in a constant state of alertness. A similar condition has already been highlighted among the general population during the COVID-19 pandemic [49,50], especially in its first months, when discussions about when and how the COVID-19 pandemic started in China were ongoing in the media, and there was a lack of understanding of COVID-19 in terms of a multitude of aspects, including from a medical point of view, issues of gaining immunity after infection, or finding a vaccine [1].

The perception of being part of a vulnerable category and consequently frightened, due to the possibility of contracting the virus, was also found in a sample of older people in the United Kingdom [51]. In line with this data, some of the participants in the present study experienced the threat of a drastic and pervasive change of life that might directly involve themselves or their loved ones. They tried to cope with the emergency by depending on others and felt threatened by the possibility of losing their support. Pre-existing relationships were maintained because, although confined, they preserved some elements of participants’ previous constructions whereas any new elements were perceived as a threat to their consolidated world.

Anxiety and threat may result in the well-documented development of post-traumatic stress disorder (PTSD) associated with the pandemic [18]. From the perspective of PCT, a person with PTSD has encountered an extreme experience that cannot be construed in relation to their other life experiences, thereby causing anxiety. This situation may often lead to the creation of a fragmented trauma-related construct subsystem that might have been validated by their traumatic experience but is not validated by the rest of their life [52]. The person might apply this outlook to other events in order to regain a meaning to the world (e.g., any situation may become a threat of potential abuse or aggression).

Participants who faced the situation with guilt experienced changes due to the pandemic as a loss of their personal roles and perceived isolation as a limitation to their active ageing [53], with consequent negative feelings such as profound sadness. This experience is common in older people due to the general condition of loss they experience in this phase of life [54]. These participants tried to reduce loneliness by watching television rather than searching for others’ company because they felt that they might have become a burden to others. This anticipation is coherent with their usual social role, which was often based on being a reference point for others rather than asking them for help and this precisely was the role that was invalidated by the pandemic because they could not help others as they were used to doing in the past, thereby making them experience guilt.

Participants used different strategies to cope with anxiety, threat and guilt. Some tried to avoid the situation by excluding unknown or threatening events from their lives, staying at home and denying any changes occurring, as well as the emotions that would accompany these events (constriction). Most of these people had already experienced a certain degree of social isolation before the pandemic and thereby had more difficulty in taking advantage of social support [55]. These participants were resigned to their condition, whereas those who used to have active lives tried to deny the risk of contagion by ignoring the restriction measures. Social relationships and opportunities offered by new technologies were also excluded to avoid changing these participants’ usual construction of themselves and the world. This solution prevented some of them from facing guilt.

In contrast, other participants faced the emergency by construing new meanings and implementing personal strategies to make the best of it. Kelly [34] referred to this transition in terms of aggression, defined as “the active elaboration of one’s perceptual field” (p. 508). This solution was the most adaptive in this situation, especially for older people: they felt better and were able to project towards the future. Previous studies have pointed out that older people who count on personal resources and have positive self-perceptions show lower reactivity to stress and higher levels of resilience [56]. In addition, these participants could confide in others by differentiating among them according to their needs.

These results underscore the well-known buffering role of social support. Close relationships usually act as emotional regulators in older people and promote a positive vision of life [57]. Nevertheless, participants differed in the benefits derived by this support. Older people who had previously experienced loneliness saw their conditions worsen during the lockdown, whereas those who already had a wide and differentiated social network benefitted from it. Other studies [51] have shown that older people’s levels of anxiety and depression have not increased, but they expressed more loneliness, especially in relation to the dimension of a social network. Bailey et al. [58] also found an increase in loneliness in older people during the lockdown in Ireland. Nevertheless, this is the first study to look deeper into older people’s experiences of loneliness and differentiate among them.

Finally, the results confirm the benefits of new media [59] when used to maintain relationships and mitigate loneliness: these tools allow older people to renew or develop social contacts and actively engage in their communities. Technology not only helps prevent older people from becoming socially isolated and lonely due to life changes including retirement, bereavement, and a deterioration in health, but it also helps those who are socially isolated escape their situation. In the context of our study, computers and mobile phones acted as a critical and transitive medium to which older people resort to as far as their loneliness is concerned, but again, participants in this study differed in their approaches and previous experiences with these means. Not all participants considered technology useful: people with age-related impairments or low digital literacy had difficulties approaching new media. Moreover, some people preferred to avoid or limit their exposure to the new elements introduced by these media especially in terms of social relations. However, others made the most of it, precisely to maintain their relationships and reconstruct them in the new forms allowed by the constraints imposed by the anti-contagion measures.

The role of television was more similar among participants—all of them agreed that it was helpful to distract themselves or to keep them company. Nevertheless, participants’ narrations showed that the great quantity of information on television and other media could also increase anxiety and feelings of threat. This result is in line with previous studies [60] that have shown that during the pandemic people sought information from numerous sources, including television (44.31%), social networks (23.45%) and newspapers (14.04%) and this search changed risk perception and the adherence to anti-contagion measures. At the same time, the spread of news from varying sources could increase misinformation [61] especially when the news was presented in a sensationalist way lacking authoritative references, or it encouraged political and economic discussions rather than medical and scientific ones, as Italian newspapers sometimes did in the first wave of the pandemic [62]. Previous studies [33,63] associated the infodemic with a decrease in psychological well-being and an increase in unhealthy risk behaviours due to a lack of understanding of the overwhelming and often contradictory information. In fact, a lack of health literacy may prevent people from understanding relevant information and taking care of their health, and this can lead to irrational behaviours that do not comply with COVID-19 policies [64].

High health literacy has already been indicated as a protective factor against depression in COVID-19 patients [65]. To promote health literacy, news must present information in a clear, transparent and accessible way [7]; at the same time, older people must be supported in distinguishing critically trustworthy information from fake news [66]. The results of our study encourage the simplification and accessibility of new technology among the older population [67].

### Limitations and Future Perspectives

This study has some limitations. First, collecting data through phone calls made fully comprehending questions difficult for hearing-impaired participants. A suggestion that can be derived from this observation for future research would be to submit a document with the list of questions before the interviews in addition to information on the study, this would allow participants to read the questions in advance or while listening. Another possibility would be to involve caregivers to help participants understand questions during the interview.

Second, the snowball sampling entailed contacting people who have common backgrounds. Most of them had social networks, even if limited, and sufficient proficiency in the use of technology and media. Although participants’ results were sufficiently heterogeneous, future research could include more people belonging to different communities (e.g., those living in residential communities, ethnic minorities, or those who are more isolated).

Finally, it is important to underscore that none of the participants experienced COVID-19 directly. Future works should gather the experiences of people who contracted COVID-19 and compare them with those presented here to reach a more comprehensive view of older people’s experiences.

## 5. Conclusions

This is the first study to collect older people’s experiences during the first wave of the COVID-19 pandemic in Italy. Moreover, this study adds to the quantitative studies [51,58] that have been conducted at the same time with older people in other countries (the UK and Ireland), with an in-depth analysis of their experience because it used semistructured interviews and a qualitative methodology for data analysis. The authors used PCT to understand and differentiate among older people’s experiences because this is a theory that considers people as the creators and experts of their worlds of meanings. Thereby, changes experienced by people are not due to external events but rather to the experience of incompatibility with their usual ways of construing events, which leads to the possibility of finding a new meaning. The COVID-19 pandemic is posing new challenges that need to be faced. PCT may offer a useful framework for understanding these changes in meaning making and helping older people cope better with the situation in daily life and in therapy and, in some cases, also recovering from psychological suffering and strain.

The different transitions identified in the results of this study suggest that the COVID-19 pandemic has impacted older people’s lives in various ways and they have faced the situation in different ways. The data showed that the pandemic changed most participants’ daily lives. In some cases, it exacerbated existing weaknesses, whereas other participants were able to take advantage of internal and external resources to cope with the situation. This condition characterized people who were not overwhelmed by the events, instead changing their own constructions of experience to adapt to the new situation.

To conclude, this study highlighted that isolation had a stronger impact on people who felt more alone or experienced a loss of role. Therefore, the likelihood of developing physical and psychological problems is not based as much on living alone but on feeling alone [68]. Older people have proven able to use their personal and social resources to cope with the situation. The more resources they have, the better they could cope. Social support, in particular, proved to be an important resource not only when received by others but also when given to others. Participants who coped better with the situation were those who considered social isolation as a way to protect others and gave meaning to the pandemic in terms of a change that the community could benefit from.

Even though this study has taken place during a particular and unprecedented time, it may represent a new contribution to ageing studies by proposing an approach focused on older people’s personal experiences of loneliness rather than their being alone, suggesting that the use of media may be a protective factor from social isolation but it must be considered again referring to the specificity of each older person’s experience, and finally offering suggestions for public health that can take into account of the people’s experiences of age-specific vulnerabilities. Institutions could promote active ageing policies aimed at helping older people who face social isolation and loneliness via their engagement in activities (such as volunteering, where and when possible) that allow them to feel connected with others and useful [13]. As the present study has shown, those who can find meaning in an emergency situation (such as the COVID-19 pandemic or any other one) and redesign their social relationships cope better with isolation and loneliness. This result can also be extended to understand older people’s experience in non-emergency situations and further studies could explore how meaning making and the active elaboration of one’s own perceptual field may help older people to experience satisfaction in life.

## Figures and Tables

**Figure 1 ijerph-18-11832-f001:**
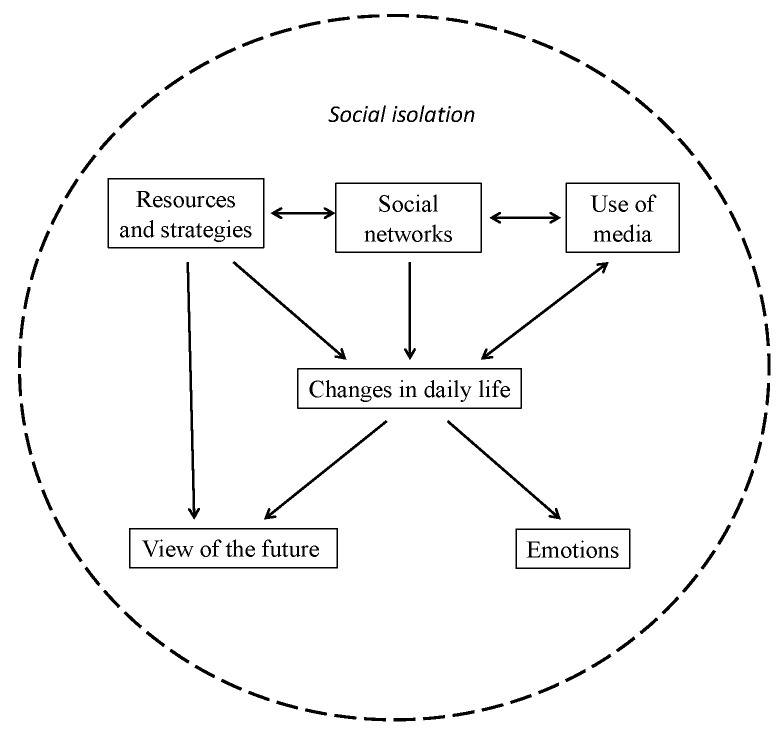
Map of the relationships between themes.

**Table 1 ijerph-18-11832-t001:** Themes and codes with the number of interviews where each code was found in parenthesis.

THEMES	CODES
Changes in daily life	Physiological and psychological changes (12) Isolation as imposition (11) Isolation as responsibility (6) No changes registered (10) Changes of habits (8) Changes as opportunity (5) Reorganization of activities (9) Changes in a spiritual sphere (8) Cessation of help in housework (3) Loss of grandparents’ role (7) Limitations to active ageing (3)
Emotions	Concern (9) Fear (6) Obsession (2) Incredulity (5) Sadness (7) Loneliness (6) Anger (2) Peacefulness (3)
Social networks	Distance from relatives (9) Distance from friends and acquaintances (14) Previous withdrawal (4) Cohabiting partner (6) Physical closeness of children (2) Occasional family visits (6) Relationship with neighbours (12) Neighbours’ withdrawal (6) No relationships maintained (6)
Exploited resources and strategies	External resources (15) Internal resources: practical and emotional issues (13) Internal resources: rediscovery of hobbies (17) To go out and skip the anti-contagion norms (4) Attempt to find the origin of the virus (2)
Use of media	Telephone contacts with relatives (12) Telephone contacts with friends (18) Telephone contacts with neighbours (2) Telephone contacts with colleagues (2) Technological skills (11) Difficulty of using technologies (15) Efficient means (9) Inefficient means (2) Proactivity in learning to use new media (4) Physical limits in using technologies (4) Lack of learning (11) Personal preferences (7) Television (12) Infodemic (8)
View of the future	Hope (3) Long-term duration (5) Long-term consequences (5)

## Data Availability

Considering this study uses narrative data, data will not be made available due to ethical and privacy restrictions.

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
