# Peer review of "Older People’s Lived Perspectives of Social Isolation during the First Wave of the COVID-19 Pandemic in Italy"

_ijerph, 2021, doi:10.3390/ijerph182211832_

Round 1

Reviewer 1 Report

The manuscript “Older People’s Lived Perspectives of Social Isolation During the First-Wave COVID-19 Pandemic in Italy” is a further contribution to the growing number of works about the psychological impact of the Covid-19 pandemic. The originality of the present study consists in analyzing qualitatively older people’s experiences of the lockdown due to the pandemic. For this reason, it provides an advance towards the current knowledge.

The authors have resorted to Kelly’s personal construct theory (PCT) to understand such experience. While I think that it is an appropriate choice, we must consider that many readers may not be familiar with it, and I wonder if the definitions of the transitions given in Table 1 allow sufficient understanding (parenthetically, the transitions are presented as diagnostic constructs in the text and professional constructs in the table, which can be confusing). Could the authors add to the original definitions a reformulation of them in simple terms?

Table 2 left me puzzled too. I cannot understand if and how the themes listed in the column on the left refer to the codes in the second column, but maybe it could be a matter of layout.

Other than that, the study is written in an appropriate way, correctly designed, and the conclusions appear supported by the results of the analysis.

Author Response

We have delated Table 1 and included in the text a more detailed presentation of the transitions with a reformulation in simpler terms and avoiding the confusion due to the interchangeable use of the terms “professional” and “diagnostic constructs”.

We have revised Table 2 because we noticed that the names of some codes probably did not allow the reader to understand how they fitted in certain themes. We have also added some space between one theme and the other in order to avoid errors due to the layout. We hope now it is clearer.

Reviewer 2 Report

This article is a very welcome contribution to the growing social science literature on people’s experiences during the pandemic. How older people experienced the dramatic changes to everyday lives and social contact that the pandemic brought with it is an especially important topic in this field. And the focus on Italy, the first European country to institute a national lockdown, is very appropriate. The methods and theoretical framework are robust and the results are interesting (if not hugely surprising). Nonetheless, I have a few suggestions for corrections, and also think that the richness of the material could be brought out a little more in the analysis through further contextualisation of people’s experiences in the particular historical moment when data was collected.

Introduction:

  • From the way it is currently written, it is not clear whether the recommendations for actions to reduce isolation laid out in the introduction are the recommendations made by the researchers whose work is cited, or are the recommendations of the authors of this article. If the former, this should be made clearer. If the latter, then providing recommendations in the introduction seem misplaced and I would recommend limiting this section to a review of the issues/literature and an overview of the research questions/contribution/approach of this study.

  • The final paragraph of the intro implies that the main focus of the article will be on the use of new and traditional media to cope with self-isolation. But I did not find that emphasis to be reflected in the thematic analysis or discussion, which seemed to be much broader in scope and to consider experiences and coping strategies more widely.

Methods:

  • Slightly confused by the opposition between ‘remaining close to lived experience’ and ‘analysing’. I would see the kind of thematic analysis undertaken as ‘analysis’ and do not see that these two things are opposed.

  • Can the authors tell us anything more about the demographics of their participants? Apart from their age, we know little about their education, income/employment, housing situation (house owner, rental, living along/with others), marital status, ethnicity, pre-existing medical conditions. This would help us to get a sense of the broad spectrum of lived experiences of the pandemic represented in the study.

Results:

  • P5 The description of how the PCT approach connects with the thematic analysis is a bit opaque in the final paragraph of the intro to this section, and similarly unclear in the sub-sections discussing the different PCT constructs. This results in the themes being a bit under-utiliised in the analysis (or at least it is not clear what part they are playing). Can you find some way of making the link to individual themes more visible in the PCT sub-sections?

Discussion:

  • The discussion would benefit from more context being provided as to how the pandemic in Italy was playing out between March-April 2020 when the interviews were conducted. E.g. , how long had the virus been detected in the country, how many cases/deaths by this point? Transmission rates? What was the media saying? What uncertainties were at play in the science around the virus? And exactly what laws/guidance were in place to govern social distancing and lockdown? It is important that the results are taken as a reflection of older people’s experiences of a particular (unprecedented in their lifetime) moment in the pandemic, rather than of ‘all’ lockdown scenarios more generally.

  • Have any similar studies been carried out in different countries/at different points in the pandemic, and how does this study compare to those?

Conclusion

  • The conclusion could do more to show what is novel/important about this study, how it builds on/is different from other research that is out there, and what new contributions it makes to the field (social psychology/Covid-19 studies/aging studies)

  • The recommendation that older people are encouraged to volunteer to overcome social isolation seems a bit inappropriate given the research is about isolation during a national lockdown, when such volunteering work might both have been illegal and to have put the participants at risk.

Author Response

We thank the reviewer for her/his positive comments about the manuscript and for the suggestions about how to improve it. We took all comments into account, and we modified the manuscript accordingly. We list the changes below as an answer to each comment and report the part of the manuscript  that has been changed. We hope the Reviewer is satisfied with our modifications.

Introduction:

  • From the way it is currently written, it is not clear whether the recommendations for actions to reduce isolation laid out in the introduction are the recommendations made by the researchers whose work is cited, or are the recommendations of the authors of this article. If the former, this should be made clearer. If the latter, then providing recommendations in the introduction seem misplaced and I would recommend limiting this section to a review of the issues/literature and an overview of the research questions/contribution/approach of this study.

Authors’ response: The recommendations for actions to reduce isolation laid out in the introduction are the recommendations made by the researchers whose work is cited. We have rephrased some sentences in order to make this more explicit.

p.2: Some researchers [31] have already suggested encouraging younger generations with higher technical skills to support older people in the use of new media. Traditional media (e.g. television and radio) can also be useful instruments to maintain connectivity with the rest of the world. However, exposure to news could cause distress and more anxiety [10], so it is recommended to regulate the use of media. It has been suggested that people should limit their time searching for information—a maximum of once or twice per day [17] and preferably not at night, to facilitate falling asleep [32].

  • The final paragraph of the intro implies that the main focus of the article will be on the use of new and traditional media to cope with self-isolation. But I did not find that emphasis to be reflected in the thematic analysis or discussion, which seemed to be much broader in scope and to consider experiences and coping strategies more widely.

Authors’ response: We agree that the thematic analysis and discussion are much broader in scope than the use of new and traditional media, nevertheless this was a specific focus that has now been more emphasized in the results and discussion.  We have also reformulated the final paragraph of the intro and the first of the abstract in order to clarify that the use of new and traditional media to cope with self-isolation was not the main focus of the study but an additional/specific one.

p.2: More specifically, the role of media and online technologies was also explored. Participants’ experiences were differentiated based on the transitions elaborated within PCT to identify personalized strategies for health promotion.

Methods:

  • Slightly confused by the opposition between ‘remaining close to lived experience’ and ‘analysing’. I would see the kind of thematic analysis undertaken as ‘analysis’ and do not see that these two things are opposed.

 Authors’ response: We agree with the reviewer’s comment and have removed this contraposition and also the previous ones as we think they are not necessary.

  • Can the authors tell us anything more about the demographics of their participants? Apart from their age, we know little about their education, income/employment, housing situation (house owner, rental, living along/with others), marital status, ethnicity, pre-existing medical conditions. This would help us to get a sense of the broad spectrum of lived experiences of the pandemic represented in the study.

Authors’ response: we have added the information required: ethnicity, employment, socio-economic status and marital status. We had already specified that they were all living at home -alone (N = 12), in a couple family unit (N = 14), or with a home health aide (N = 2)-. We do not have information about education and housing situation (house owner, rental) because we did not want to investigate these aspects to avoid being perceived as too much investigative.

P.3: The sample analysis is thus composed of 28 participants (10 M, 18 F) from 72 to 94 years old (mean = 80, SD = 5.742), all Caucasian and retired with a low (N = 12), middle (N = 5) or high (N = 11) socio-economic status, married (N=14), widowed (N=11), or unmarried (N=3), and living at their home alone (N = 12), in a two-person family unit (N = 14), or with a home health aide (N = 2).

Results:

  • P5 The description of how the PCT approach connects with the thematic analysis is a bit opaque in the final paragraph of the intro to this section, and similarly unclear in the sub-sections discussing the different PCT constructs. This results in the themes being a bit under-utiliised in the analysis (or at least it is not clear what part they are playing). Can you find some way of making the link to individual themes more visible in the PCT sub-sections?

Authors’ response: We have made themes more evident in the PCT sub-sections by specifying to which themes the reported experiences referred to and when the themes or the codes attributed to a specific experience characterizing a certain transition are not explicitly mentioned in the text, they are reported in parentheses. We have specified this in the paper at p.10 before presenting the results corresponding to each transition.

The whole result section has been re-written according to this direction.

p.5 Each transition is described through the presentation of the experiences grouped together on each theme: changes in daily life, emotions, social networks, exploited resources and strategies, use of media, and view of the future. When the themes or the codes attributed to a specific experience characterizing a certain transition are not explicitly mentioned in the text, they will be reported in parentheses.

3.1 Anxiety

Anxiety was the main transition experienced when participants found it difficult to give meaning to the pandemic, which turned their lives upside down to the point of making them unrecognizable. Changes implied by the pandemic were described as unreal and belonging to another world. The narratives themselves were often difficult to follow and seemed chaotic to the interviewer: “I never expected to live something like this [the pandemic] into my life” (P3, F, 73); “I seem to live in another world” (P10, F, 88); “In certain moments it looks like if I was on another planet, I seem to dream because I say to myself: ‘can it be possible a thing like this?’ […] Sometimes I don’t know if it is reality or fantasy.” (P26, F, 94)

These participants’ prevalent emotions were incredulity about the spread of the virus and the fear of confronting something unknown and invisible. This feeling also escalated to panic and obsession: “When you come home, you never know if it [the virus] is there. […] When I come home, I start to disinfect myself, wash myself, disinfect everything, everything that comes into the house and so [there] is a little bit of an obsession.” (P3, F, 73).

Not being able to give meaning to their own experience, interviewees implemented strategies that proved to be inefficient in coping with anxiety, or they looked for a possible origin of the virus in supernatural entities (exploited resources and strategies): “I can’t read, I can’t sew, because I can’t see [what this is]. This is not living.” (P8, F, 83); “I pray three times a day against this thing that is all around. […] Because this thing [the corona virus] comes from Satan […] Who punishes us is Satan.” (P27, F, 84).

Narrations showed that these participants passed from the panic of going out of their home to going around to escape from loneliness: “I don’t go inside the houses, but while going through the street I go to ring the bell and I say, ‘Hello, how are you? Is it all right?’“ (P8, F, 74). One participant became suspicious about other people who were viewed as possible vehicles of infection and felt angry at people who denied the situation: “When I see some people who are not taking it seriously, they scare me, they make me [so] angry that you have no idea.” (P3, F, 73).

These participants were alone or had wide undifferentiated social networks. Other people’s behaviours were described as unknown and unpredictable: “I do not know when the path will be given to Liberation, that is when people will be able to go out, here and there, what might happen … because people stay at home. Yes, they do but also, they do not … my sister does not respond to the phone, does not hear, does not respond. To be honest, I don’t even know where she lives.” (P27, F, 84)

Sometimes participants maintained their pre-existing relationships through the telephone but this could become an additional source of anxiety, either due the difficulty in using it or because of receiving a large number of WhatsApp messages: “[How to use the smartphone] is all to be learned but there is not much passion to learn it, this is the trouble, because I feel powerless and I am tempted to throw it away […] We [I and my friends] message each other even too much because […] you open your cell phone and find 110 messages, I say ‘it is not possible’” (P3, F, 73). Television also might become a source of anxiety for the overwhelming news transmitted: “When they [media] show you how many people are dead, how many of these or those [things], I said […] ‘Stop now otherwise you go out of your mind’” (P8, F, 83).

Views of the future were vague, and participants pointed out the difficulty of anticipating it: “I don’t know how this is going, will we get ill too? I don’t know.” (P7, F, 74).

3.2 Threat

Some participants were mainly threatened by the possibility that they or their loved ones might contract the virus: “This period has begun with a certain fright […] Also because my husband and I are of an age to be at higher risk.” (P5, F, 74). Changes undermining their certainties were perceived as pervasive and involved the physical domain in terms of sleep, appetite, and attention disorders: “What is the problem? I don’t eat at the right times at home, and I don’t go out but I’m always deadly hungry.” (P9, M, 83). Concern was the prevalent emotion, and led these participants to stay at home and rigorously follow anti-contagion measures with the consequent limitations in their daily life and social relationships because others were seen as possible vehicles of contagion: “ I respect the rules I have received” (P12, F, 74); “I always go out with the mask on, gloves and mask.” (P4, M, 89); “My son lives here, in the same floor […] but we don’t see each other because they go out. Maybe they bring something home.” (P7, F, 74).

In terms of social networks, these participants usually had few strong social ties (mainly within the family) and used to rely on them to solve practical problems or emotional issues: "It is [our daughter] who brings us our groceries home. [...] She brings us our groceries, she leaves them in the street in a basket." (P9, M, 83); “Luckily, I have my husband, and so we exchange a few words.” (P22, F, 80). They kept in touch with neighbours if there was already a pre-existing relationship; in this case, people with bordering backyards felt they had an advantage because they could keep in contact with others without being too far away from their home: “You stay in the garden and you have a chat [...] my contacts are only with my neighbours and we cannot do anything else, when I go out with my dog I always meet the same people I met before with their dogs, and we say goodbye, we have no contact.” (P21, F, 77) “We are all inside our houses. Nobody goes out, nobody speaks to each other, no one opens the balcony window to allow the air in. You see nobody around and I have no relationship with anyone because most of my neighbours are new. I had a good relationship with the lovely people who lived here before, but they are not here anymore. Therefore, I don’t know these new ones.” (P20, M, 84).

The only medium used to keep in contact with a close circle of relationships, yet also at a distance, was the telephone, whereas using new media was felt to be tiring especially as it was considered something new that might imply a change in the participants’ comfort zone: "Unfortunately I struggle to use those very modern things... I have only the telephone. I only make and receive calls." (P4, M, 89). Due to worries about the pandemic, the participants watched television to keep themselves “informed” (P6, F, 74) but exposure to lots of bad news ended up becoming a source of additional worry due to the huge amount of alarming news broadcast: “I watch the television to catch up on the pandemic.” (P20, M, 84); "I watch the television, what they say, but not too much because otherwise, it makes you sick." (P24, F, 72).

Finally, threat involved participants’ views of the future that were characterized by concerns about the psychological consequences of isolation: “The sooner [the isolation] ends, the better. Otherwise, if it goes on, they will have not only the virus but also depression.” (P22, F, 79).

3.4 Guilt

Due to the pandemic, ten participants experienced changes with a loss of their personal roles (guilt) and felt a gap between their previous lives and present conditions. These participants were used to mainly looking after themselves and taking care of others (not just family members but also friends and other people in the community) rather than asking others for help. The pandemic implied profound changes in these core aspects of their lives. In particular, participants complained about the loss of their role as grandparents: “I took care of my grandchildren, I took them, I brought them, and so on.” (P1, F, 78). Moreover, narrations showed that participants remained at home as much as possible because they understood the importance of norms and felt responsible for others’ safety: "I am duty-bound, that is, if they impose something on me, I think about it and I do it. [... ] I am very responsible and so is my husband." (P12, F, 74). 

Although acknowledging the existence of the virus and understanding its risk, three participants considered isolation a limitation that implied a loss in their activity (e.g. travelling, physical activity) while ageing: “I was and am a person who goes out at 9 am and comes back at 1 pm; who goes out again at 4-4.30 pm and comes back at 8 pm. Home makes me feel nostalgic […] I feel stuck in writing, stuck with my whole nervous system. I always feel nervous, alone […] Before, one was free; now we are [in a] home detention agreement without having committed any crime.” (P13, M, 79).

The deprivation of physical contact, limitations in their lives and in the possibility of taking care of others implied a loss of meaning and caused feelings of loneliness and sadness, which were the prevalent emotions expressed in these participants’ narratives: “I miss physical contact with other people who we now only talk to by phone. […] I have to hold them [my grandchildren] off, this is a sorrow.” (P12, F, 74); “I hear the bells but I can’t be in the church, because the churches are all closed. I need this and I want to die.” (P26, F, 94); “I call my daughter very little because she works at the hospital. [...] She hardly calls me, I am reluctant to disturb her.” (P1, F, 78).

Some of these participants watched television to cope with loneliness, because it was an already widely used strategy to distract themselves without being a burden to others. At the same time, someone emphasized the spread of information linked to the pandemic, which was usually expressed in a crude and repetitive way: “Many people said to me that they switch off the television because it makes them crazy, they said that it conveys terrible anxiety, distress, and depression.” (P1, F, 78).

These participants did not often speak of the future, but when they did, it was in a pessimistic way, heightening the feeling that nothing would be as it was before: “If we will get out, when we will get out, I don’t know, and how we will get out, but we will be a bit suspicious of each other.” (P3, F, 73).

3.3 Constriction

In the face of anxiety, threat, or guilt seven participants tried to maintain or regain their usual construction of themselves and the world by narrowing their perceptive field (constriction) and excluding the new elements introduced by the unprecedented situation represented by the COVID-19 pandemic. Some of these participants did not recognize any changes between their present and previous life: most were already used to staying at home due to age-related health problems or caregiving roles. In one case, a method to continuing their previous way of life was to find subterfuges to dismiss the restriction norms: “When I was younger, I liked to go out. Now I stay quiet because […] I have a lot of illnesses.” (P2, F, 88); “I have to take care of my ill husband, so I’m still at home.” (P15, F, 81); “I have to sneak around to go into the woods and not be seen.” (P14, M, 76).

Constriction resulted in a lack of expressed emotions and the attempt to avoid answering questions regarding feelings associated with the current situation: “I am not living badly, eh, because, you know, at my age, it’s not that it is so hard to stay at home.” (P2, F, 88); “I live well enough that I have no particular problems. […] I do what is allowed, for example, the groceries. I just do this. Do you have any specific questions?” (P11, M, 81).

In contrast, some participants stressed their condition of loneliness, which often continued from pre-pandemic times, as they did not have many active social relationships: “I love staying alone” (P14, M, 76); "My wife died. [...] Now this disease, this virus, it happened just a month ago when I fell down and I broke my rib and I had to stay home for 30 days all alone, always, I repeat—all alone.” (P28, M, 86).

Social isolation was increased by the lack of using new technologies, which in turn was due to a lack of skills or the will to use them to keep in contact with others: “I have never had contact with anyone. Not with the telephone, or with other media.” (P28, M, 86). Age-related impairments could make it hard to learn new applications, such as sending a message, or reading it if it had a small font size, or remembering a telephone number by heart: “I have a problem with my eyes and I see very little [...] if my friends send me a message, I don’t see it.” (P16, F, 88).

These limitations provided a sense of frustration, but participants avoided this feeling by attributing their sense of inadequacy to the technology. In addition, one participant who had good skills still considered these media useless because they lacked all the aspects of face-to-face communication: “They aren’t like real contacts. When you talk with someone and you look them in the face you can see their reactions; when you talk with someone on the computer, you don’t have this chance.” (P11, M, 81).

The narrowing of these participants’ life perspectives also resulted in a lack of future perspectives, expressed by the absence of any references to the future.

3.5 Aggression

Eight participants were not overwhelmed by the events and instead changed their own constructions to adapt to the new situation (aggression). These people reorganized their everyday lives around allowed activities: they found new solutions to cope with the situation (e.g., one person started making protective masks and another listened to the mass on television) or took isolation as an opportunity to carry on or rediscover their hobbies and passions or discover new ones and reported positive feelings associated with these activities: “I live in a very peaceful way. […] I’ve discovered the pleasure of reading.” (P19, M, 80).

They reported that feeling committed helped them to cope with loneliness and other negative emotions: “I spend the afternoon praying, then I turn on the record player and listen to music. […] Then I do crossword puzzles or if it’s sunny, I stay in the garden. […] I try to fill the day with these activities.” (P6, F, 74); “I also have a lot to do at home. I do all the jobs I have never done because I had no time.” (P7, F, 74); “The cleaning lady no longer comes since it began. […] So I have to provide for myself.” (P20, M, 84).

Some participants also attributed their positive feelings to how their character complemented the ongoing situation: “In a certain sense, this is a strange thing because I have always been a lover of peacefulness and to see the streets so empty gives me a sensation of -bah, sometimes not seeing crowds of people makes me feel better” (P9, M, 83).

Although they suffered from being distanced from loved ones (social networks) and with limitations in their daily activities, these participants respected the restrictions because they considered them necessary to protect the community’s health. They were usually attentive to other people’s needs and to the common well-being: “In this period, I am obviously at home; I go out very little, only once every 15 days for the groceries.” (P17, M, 81); “I wish that […] we can take the chance to change. I have lived most of my life but I think about the world that we’ll leave to younger people. We have to leave them a different world to the one we live in now, which is one we have broken (P5, F, 73).

In general, they autonomously provided for their needs but could also rely on others (community services, home health aides, and children) for help: “I went to my car the other night and I found [the battery] dead […] Then a gentleman who lives nearby came into my courtyard with his car and we have worked it all.” (P9, M, 83).

These participants maintained their relationships with relatives, friends, neighbours, and colleagues through the use of technology: “Luckily, during the rest of the day, I can have indirect, but useful, contact through the smartphone with both children and grandchildren.” (P20, M, 84).

These tools were considered a good alternative to the usual modalities of maintaining relationships. One person was already skilled in the use of new technologies because of personal interest or previous use at work, and was used to maintaining contact with others through them: “I keep in touch with others through the telephone or an email. [...] I used them even before, when in 1995, the Internet came out and it seemed a positive thing.” (P18, M, 76).

Other participants, while starting from low levels of competence, were motivated by the situation to rely on technology and took the opportunity to improve their ability: “For Christmas, I’ve received a smartphone, and so I have a way to send messages, to send videos, send beautiful things to hear that come to me every day. Wonderful things to watch come to me. And this helps me a lot.” (P24, F, 72).

Some of these participants looked at the end of the pandemic as an event able to modify people’s moral values and to make humanity better (view of the future); they compared the pandemic with living through the world war, after which every little thing was appreciated: “I hope, I say I hope, that when this emergency is over, people have a different way of seeing life. That it is a sign, a sign for how you lived before and after, as [during] the post-war.” (P6, F, 74).

Discussion:

  • The discussion would benefit from more context being provided as to how the pandemic in Italy was playing out between March-April 2020 when the interviews were conducted. E.g. , how long had the virus been detected in the country, how many cases/deaths by this point? Transmission rates? What was the media saying? What uncertainties were at play in the science around the virus? And exactly what laws/guidance were in place to govern social distancing and lockdown? It is important that the results are taken as a reflection of older people’s experiences of a particular (unprecedented in their lifetime) moment in the pandemic, rather than of ‘all’ lockdown scenarios more generally.

 Authors’ response: we have added the information required to give a context to the results of the study and also re-written some parts of the discussion in order to refer more precisely to the specific context participants were living.

p.9: This study aimed to explore how older people experienced and coped with isolation during the lockdown in the first Western area affected by COVID-19. The results reflected the particular context in which they were collected. In February 2020, the first cases of people impacted by the COVID-19 were registered in Italy. By the time data collection had begun (i.e. March 26th), there were 80,593 total cases and 8,241 deaths in this Country [45] and the population had been under a complete lockdown for 20 days. People had to remain at home and to wear masks and gloves at the supermarket, and it was strongly recommended not to go to the supermarket more than twice a month or once a week. The present study involved older people living in Veneto (an area of Northern Italy), where the containing measures were implemented earlier and in a harder way than in any other Western country [46] as this region registered one of the highest numbers of infections (6,935 total cases on March 26th) and one of the highest numbers of bed occupancy in Intensive Care Units (326) [47].

The results of the present study pointed out a variety of responses in terms of day-to-day life changes, emotions felt, social networks, exploited resources, and the use of technology. Previous studies have already underlined that older people form a large and heterogeneous group; for example, they differ in terms of genetics, life and cultural experiences, health, and lifestyle [48]. To better describe the heterogeneity of the answers collected and identify common trajectories, the researchers referred to the main transitions in terms of PCT that participants experienced concerning the changes brought by the COVID-19 pandemic: anxiety, threat, guilt, constriction, and aggression.

The awareness that one can construe, only partially, events that are on the border of the range of convenience of one’s construct system is what Kelly [34] referred to as anxiety. In line with previous studies [5,16], anxiety in particular was the prevalent experience for some participants in this study who spoke about the pandemic mainly in terms of something invisible and unknown that upset their lives. They could not find an effective way to cope with the situation. Instead they resorted to superstitious solutions and became suspicious about others and the outside world, thereby living in a constant state of alert. A similar condition has already been highlighted among the general population during the COVID-19 pandemic [49,50], especially in its first months, when discussions about when and how the COVID-19 pandemic started in China were ongoing in the media, and there was a lack of understanding of COVID-19 in terms of a multitude of aspects, including from a medical point of view, issues of gaining immunity after infection, or finding a vaccine [1].

The perception of being part of a vulnerable category and consequently frightened, due to the possibility of contracting the virus, was also found in a sample of older people in the United Kingdom [51]. In line with this data, some of the participants in the present study experienced the threat of a drastic and pervasive change of life that might directly involve themselves or their loved ones. They tried to cope with the emergency by depending on others and felt threatened by the possibility of losing their support. Pre-existing relationships were maintained because, although confined, they preserved some elements of participants’ previous constructions whereas any new elements were perceived as a threat to their consolidated world.

Anxiety and threat may result in the well-documented development of post-traumatic stress disorder (PTSD) associated with the pandemic [18]. From the perspective of PCT, a person with PTSD has encountered an extreme experience that cannot be construed in relation to their other life experiences, thereby causing anxiety. This situation may often lead to the creation of a fragmented trauma-related construct subsystem that might have been validated by their traumatic experience, but is not validated by the rest of their life [52]. The person might apply this outlook to other events in order to regain a meaning to the world (e.g., any situation may become a threat of potential abuse or aggression).

Participants who faced the situation with guilt experienced changes due to the pandemic as a loss of their personal roles and perceived isolation as a limitation to their active ageing [53], with consequent negative feelings such as profound sadness. This experience is common in older people due to the general condition of loss they experience in this phase of life [54]. These participants tried to reduce loneliness by watching television rather than searching for others’ company because they felt that they might have become a burden to others. This anticipation is coherent with their usual social role, which was often based on being a reference point for others rather than asking them for help and this precisely was the role that was invalidated by the pandemic because they could not help others as they were used to doing in the past, thereby making them experience guilt.

  • Have any similar studies been carried out in different countries/at different points in the pandemic, and how does this study compare to those?

 Authors’ response: We have added the reference to some studies that have been published in the meantime on older people’s health and loneliness during the pandemic and have discussed the results at the light of these studies.

As an exemple p.11: Other studies [51] have shown that older people’s levels of anxiety and depression have not increased, but they expressed more loneliness, especially in relation to the dimension of a social network. Bailey et al. [58] also found an increase in loneliness in older people during the lockdown in Ireland. Nevertheless, this is the first study to look deeper into older people’s experiences of loneliness and differentiate among them.

Conclusion

  • The conclusion could do more to show what is novel/important about this study, how it builds on/is different from other research that is out there, and what new contributions it makes to the field (social psychology/Covid-19 studies/aging studies)

 Authors’ response: We have better highlighted what this study adds to the previous ones.

p.12: This is the first study to collect older people’s experiences during the first wave of the COVID_19 pandemic in Italy. Moreover, this study adds to the other studies conducted with older people in other countries (the UK and Ireland), with an in-depth analysis of their experience that allows differentiating among them. The authors used PCT to achieve this objective.

  • The recommendation that older people are encouraged to volunteer to overcome social isolation seems a bit inappropriate given the research is about isolation during a national lockdown, when such volunteering work might both have been illegal and to have put the participants at risk.

Authors’ response: We have better explained and contextualized this suggestion.

p.12: Even though this study has taken place during a particular and unprecedented time, it may represent a new contribution to ageing studies and offer useful suggestions for public health. Institutions could promote active ageing policies aimed at helping older people who face social isolation and loneliness via their engagement in activities (such as volunteering, where and when possible) that allow them to feel connected with others and useful [13]. As the present study has shown, those who can find meaning in an emergency situation (such as the Covid-19 pandemic) and redesign their social relationships cope better with isolation and loneliness.

Reviewer 3 Report

The sample size is too small. 

Author Response

The sample size follows the criteria of qualitative research which states that sampling ends once theoretical saturation is reached, that is the point at which gathering more data does not lead to more information related to the research questions (Flick, 2009), usually when the number of interviews reaches around 15 (+/-10) (Kvale, 1996). We have now added this in the text.

Reviewer 4 Report

The topic developed in the article Older People’s Lived Perspectives of Social Isolation During 2 the First-Wave COVID-19 Pandemic in Italy, is very current and relevant. The only observation I can make is regarding the sample of participants selected for the study, which is not very significant. However, the methodological treatment and the results obtained allow us to make inferences about what is happening worldwide.

Author Response

We thank the reviewer for his/her comment. We agree that the sample is very specific of a geographic area and we have included this comment among the study limitations, nevertheless we appreciate that the reviewer noticed that the results may be usuful to understand older people’s experience also in other parts of the world.

Round 2

Reviewer 2 Report

The authors have taken the time to consider each of the points raised in the review carefully and I found the article to be greatly improved. In particular, the scope of the study and its contribution were clarified, the connections between the thematic analysis and discussion sections have been strengthened, and there is better sign-posting to the reader in general. 

I still found the conclusion to be a little weak by comparison with the rest of the paper. In particular, the newly added sentences at the beginning of the section feel slightly formulaic and still fail to scale-up from this specific study to consider its broader implications (e.g. for how we understand aging and isolation more generally, making a case for PCT as a method that could be applied to public health emergencies more widely). As it stands, these sentences read: "This is the first study to collect older people’s experiences during the first wave of the COVID_19 508 pandemic in Italy. Moreover, this study adds to the other studies conducted with older people in other countries (the UK and Ireland), with an in-depth analysis of their experience that allows differentiating among them. The authors used PCT to achieve this objective."

I find the second sentence a bit clunky. What makes this research in-depth by comparison with the other studies cited? Why is it so important to differentiate between experiences and how does this contribute to understandings of aging/isolation/pandemics more broadly? The third sentence mentions the use of PCT, but doesn't really make a case for the use of PCT as an original/novel contribution to scholarship in this area.

Further on in the conclusion, the writing style is similarly underwhelming. e.g. to say that the pandemic impacted people's lives in 'various ways' seems to be stating the obvious. What is specific to the findings of this study that we should be paying attention to?

The final paragraph of the conclusion states that 'Even though this study has taken place during a particular and unprecedented time, it may represent a new contribution to ageing studies and offer useful suggestions for public health.' But what this contribution to ageing studies is, is not spelt out here. e.g. is it about showing the value of a PCT approach? About what it reveals about media/isolation interactions? About the differences between being alone and loneliness? Moreover, rather that providing a disclaimer about the research taking place during a public health emergency, surely one of its contributions is to demonstrate the importance of research on ageing in relation to public health emergencies of all kinds, precisely because of people's experiences of age-specific vulnerabilities? 

In other words, I think the conclusion could still be made a little bolder, with the novel contributions of the study spelt out a little more clearly so that readers know what to take from it. 

Author Response

We thank the reviewer for her/his recognitions of the improvement of the manuscript and for the further suggestions. We took them into account, and we modified the manuscript accordingly. We list the changes below as an answer to each comment and report the part of the manuscript that has been changed. We hope the Reviewer is satisfied with our modifications and will consider the manuscript suitable for publication in its current form.

I still found the conclusion to be a little weak by comparison with the rest of the paper. In particular, the newly added sentences at the beginning of the section feel slightly formulaic and still fail to scale-up from this specific study to consider its broader implications (e.g. for how we understand aging and isolation more generally, making a case for PCT as a method that could be applied to public health emergencies more widely). As it stands, these sentences read: "This is the first study to collect older people’s experiences during the first wave of the COVID_19 508 pandemic in Italy. Moreover, this study adds to the other studies conducted with older people in other countries (the UK and Ireland), with an in-depth analysis of their experience that allows differentiating among them. The authors used PCT to achieve this objective."

I find the second sentence a bit clunky. What makes this research in-depth by comparison with the other studies cited? Why is it so important to differentiate between experiences and how does this contribute to understandings of aging/isolation/pandemics more broadly? The third sentence mentions the use of PCT, but doesn't really make a case for the use of PCT as an original/novel contribution to scholarship in this area.

Authors’ response: We thank the reviewer for this suggestion and tried to clarify more what makes this research in-depth by comparison with the other studies cited and why it is important to differentiate between experiences and how this contributes to understandings of aging/isolation/pandemics more broadly. We also specified why the use of PCT may represent an original/novel contribution to scholarship in this area.

p.12 This is the first study to collect older people’s experiences during the first wave of the COVID_19 pandemic in Italy. Moreover, this study adds to the quantitative studies [51,58] that have been conducted at the same time with older people in other countries (the UK and Ireland), with an in-depth analysis of their experience because it used semi structured interviews and a qualitative methodology for data analysis. The authors used PCT to understand and differentiate among older people’s experience because this is a theory that consider people as the creators and experts of their world of meanings. Thereby, changes experienced by people are not due to external events but rather by the experience of incompatibility with their usual ways of construing events, a thing that leads to the possibility of giving new meaning. COVID-19 pandemic is posing new challenges that need new meanings to be faced. PCT may offer a useful framework for understanding these changes in meaning making and helping older people coping better with the situation in daily life and in therapy and, in some cases, also recovering from psychological suffering and strain.

Further on in the conclusion, the writing style is similarly underwhelming. e.g. to say that the pandemic impacted people's lives in 'various ways' seems to be stating the obvious. What is specific to the findings of this study that we should be paying attention to?

Authors’ response: The statement “the pandemic impacted people's lives in various ways” referred to the trajectories we identified and we have now specified this. We have already presented in the discussion section what we should pay attention to and we would not like to repeat this here in detail. Here we only summarize the results discussed in the previous section.

p.12 The different transitions identified in the results of this study suggest that the COVID-19 pandemic has impacted older people’s lives in various ways and they have faced the situation in different ways. Data showed that the pandemic changed most participants’ daily lives. In some cases, it exacerbated existing weaknesses, whereas other participants were able to take advantage of internal and external resources to cope with the situation. This condition characterized people who were not overwhelmed by the events, instead changing their own constructions of experience to adapt to the new situation.

To conclude, this study highlighted that isolation had a stronger impact on people who felt more alone or experienced a loss of role. Therefore, the likelihood of developing physical and psychological problems is not based as much on living alone but on feeling alone [68]. Older people also proved able to use their personal and social resources to cope with the situation. The more resources they had, the better they could cope. Social support, in particular, proved to be an important resource not only when received by others but also when given to others. Participants who coped better with the situation were those who considered social isolation as a way to protect others and gave meaning to the pandemic in terms of a change that the community could benefit from.

The final paragraph of the conclusion states that 'Even though this study has taken place during a particular and unprecedented time, it may represent a new contribution to ageing studies and offer useful suggestions for public health.' But what this contribution to ageing studies is, is not spelt out here. e.g. is it about showing the value of a PCT approach? About what it reveals about media/isolation interactions? About the differences between being alone and loneliness? Moreover, rather that providing a disclaimer about the research taking place during a public health emergency, surely one of its contributions is to demonstrate the importance of research on ageing in relation to public health emergencies of all kinds, precisely because of people's experiences of age-specific vulnerabilities? 

Authors’ response: We have specified the contribution of this study to ageing studies and underlined how its results may be extended to research on ageing and not only in relation to public health emergencies .

p.12 Even though this study has taken place during a particular and unprecedented time, it may represent a new contribution to ageing studies by proposing an approach focused on older people’s personal experience of loneliness rather than their being alone, suggesting that the use of media may be a protective factor from social isolation but it must be considered again referring to the specificity of each older person’s experience, and finally offering suggestions for public health that can take into account of the people's experiences of age-specific vulnerabilities. Institutions could promote active ageing policies aimed at helping older people who face social isolation and loneliness via their engagement in activities (such as volunteering, where and when possible) that allow them to feel connected with others and useful [13]. As the present study has shown, those who can find meaning in an emergency situation (such as the Covid-19 pandemic or any other one) and redesign their social relationships cope better with isolation and loneliness. This result can also be extended to understand older people’s experience in non-emergency situations and further studies could explore how meaning making and the active elaboration of one’s own perceptual field may help older people to experience satisfaction in life.

In other words, I think the conclusion could still be made a little bolder, with the novel contributions of the study spelt out a little more clearly so that readers know what to take from it. 

Authors’ response: We hope that we have fulfilled this request now.

Reviewer 3 Report

Thanks, it is true that collecting more data may not lead to more information related to the research questions when the number of interviews reaches that number

Author Response

We thank the reviewer. No further changes required here.